# Overview of Scanner Invariant Representations

**Author(s) names withheld**                                            EMAIL(S) WITHHELD

## Abstract

Pooled imaging data from multiple sources is subject to bias from each source. Studies that do not correct for these scanner/site biases at best lose statistical power, and at worst leave spurious correlations in their data. Estimation of the bias effects is non-trivial due to the paucity of data with correspondence across sites, so called "traveling phantom" data, which is expensive to collect. Nevertheless, numerous solutions leveraging direct correspondence have been proposed. In contrast to this, Moyer et al. (2019) proposes an unsupervised solution using invariant representations, one which does not require correspondence and thus does not require paired images. By leveraging the data processing inequality, an invariant representation can then be used to create an image reconstruction that is uninformative of its original source, yet still faithful to the underlying structure. In the present abstract we provide an overview of this method.

**Keywords:** Harmonization, diffusion MRI, Invariant Representation

## 1. Introduction and Summary

In magnetic resonance imaging (MRI), variations in observational conditions, protocol, and equipment induce site-wise and scanner-wise biases in the collected data (Chen et al., 2014; Fortin et al., 2017; Jovicich et al., 2006). Without correcting for these biases, multi-site studies will at best lose statistical power, and in some cases may arrive at erroneous conclusions. It is therefore imperative in multi-site studies to make these corrections; the process of doing so, removing or compensating for unwanted scanner/site-wise variations, is known as harmonization. In the present work we focus on harmonization for diffusion MRI (dMRI), a modality known to have scanner/site biases (Correia et al., 2009; Giannelli et al., 2010; Pagani et al., 2010; Papinutto et al., 2013; Vollmar et al., 2010; White et al., 2009; Zhan et al., 2010, 2013, 2012) as well as several extra possible degrees of freedom with respect to protocol (e.g., angular resolution, $b$-values, gradient waveform choice, etc.).

Previous work has largely focused on the summary statistic level (e.g. Fractional Anisotropy) (Fortin et al., 2017; Zavaliangos-Petropulu et al., 2018) or on supervised cases where pairs of images from the same subject collected with different scanners are provided (Blumberg et al., 2018; Tanno et al., 2017). These methods attempt to estimate the relative effects of each scanner/site, either in derived measures or in the original data domain.

Moyer et al. (Moyer et al., 2019) present an unsupervised method that instead learns a representation of the images that is uninformed of the scanner/site at which they were collected, yet also one that is otherwise maximally informative of the image. Reconstructions from this uninformed representation will then be uninformed of their original scanner/site context, a result which follows from data processing inequality.

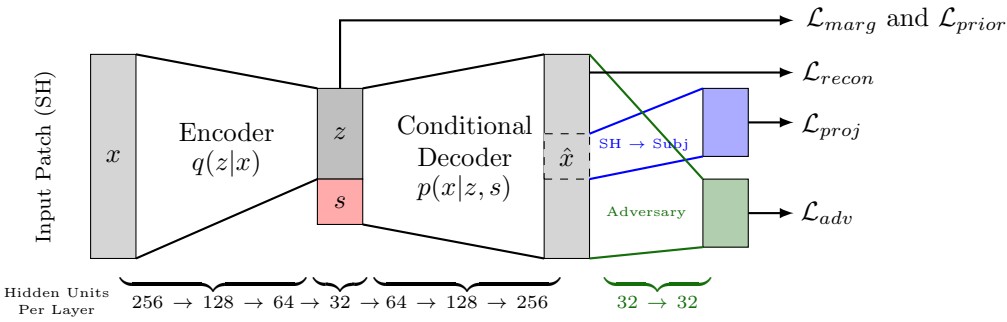

Figure 1: Diagram describing network configuration.

The construction of invariant representations is, generally, non-trivial. A previous paper by Moyer et al. (Moyer et al., 2018) show that this can be done by compressed conditional auto-encoder, where the learned encoding becomes uninformed of the conditional factor under compression. This leads to the following procedure:

1. Construct an auto-encoder for image data, using compressive regularization (e.g. penalizing $I(x, z)$ for data $x$ and encoding $z$), and condition the output on the site.

2. Train the auto-encoder on images from each scanner/site independently.

3. At test time, manipulate the conditional decoder to remap images through the learned invariant code to a single scanner/site context.

As described in (Moyer et al., 2018), conditional architectures such as the one described here (Fig. 1) penalize learned site information $I(z, s)$. In reducing site information in $z$ this architecture reduces $I(\hat{x}, z)$, the site information in the image reconstruction.

Due to hardware limitations, in (Moyer et al., 2019) this was done patch-wise. In order to improve performance the authors further included an adversary on the patch-wise output. To generalize representations across particular gradient directions, a spherical harmonics representation was used at the patch level, which was projected back to subject specific directions when calculating reconstruction loss. The proposed auto-encoders may be learned either for the two-site case ("single-task") or the multi-site case ("multi-task").

## 2. Overview of the Empirical Evaluation

Moyer et al. (Moyer et al., 2019) present an evaluation of this method on the 2018 CDMRI Challenge Dataset (Ning et al., 2018; Tax et al., 2019, 2018), which is composed of images from 15 subjects. As described in (Ning et al., 2018) data from each subject were collected on two different scanners: a 3 T GE Excite-HD "Connectom" and a 3 T Siemens Prisma scanner. For each scanner, two separate protocols were collected, one of which matches between the scanners at a low resolution (2.4 mm isotropic, two shells of $b = 1200, 3000$ with 30 gradient directions), and another which does not match at a high resolution (1.5 mm and 1.2mm isotropic, same shells 60 gradient directions). This creates a total of 4 scanner/sites (denoted P30,P60,C30,C60), for which the authors create 6 tasks, mapping to and from P30 from each of the other sites.

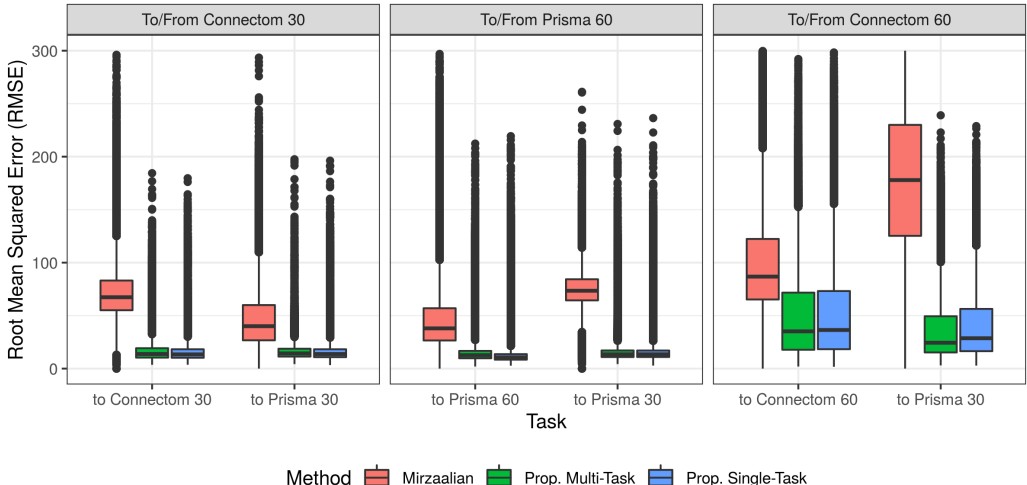

Post-hoc Adversarial Accuracy, Predicting $s$ from $z$

|  | Oracle | Full Model | No $\mathcal{L}_{marg}$ | No $\mathcal{L}_{marg}$ or $\mathcal{L}_{prior}$ |
|---|---|---|---|---|
| Proposed Single-task, C30 | 0.5 | 0.61 | 0.63 | 0.63 |
| Proposed Single-task, P60 | 0.5 | 0.5 | 0.51 | 0.54 |
| Proposed Single-task, C60 | 0.5 | 0.63 | 0.68 | 0.85 |
| Proposed Multi-task | 0.25 | 0.41 | 0.41 | 0.62 |

Figure 2: At **top** we report the predictive accuracy for each of the methods, measured by RMSE per voxel in the original data domain, for each of the six tasks. At **bottom** we report the mean test accuracy for an adversary trained post-hoc to predict the site variable $s$ from encoding $z$, and adversarial accuracy for each ablation test.

The authors split this dataset into 9 training subjects, 1 validation subject, and 5 held out-test subjects. One baseline comparison was found in the literature and compared to, Mirzaalian et al (Mirzaalian et al., 2018), which relies on a template based solution. For the proposed method, a post-hoc adversary was fit to the learned code, as a lower-bound proxy (Moyer et al., 2018) for remaining mutual information $I(z, s)$; a further ablation test for each network component was conducted. As show in Figure 2, the proposed methods significantly reduce predictive error, while also removing site information, as demonstrated by the post-hoc adversary.

## 3. Discussion and Limitations

Moyer et al. (Moyer et al., 2019) presents an alternative method for harmonization, one using intermediate scanner invariant representations to remove scanner/site information. This has theoretical advantages, yet also at least one limitation: inclusion of a low information site (e.g. a bad image quality site) will induce a low information invariant encoding $z$, as the shared space is limited to the original information. Still, the proposed methods produces quality results, while removing site information from the intermediate encoding.

## Acknowledgments

Acknowledgments withheld.

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
