# OpenReview forum: "Overview of Scanner Invariant Representations"
_MIDL.io/2020/Conference — MIDL 2020_

### Official Review · AnonReviewer2 · 2020-02-27
**Review of approach by Moyer at al  on correcting for scanner differences**

**Rating:** 2
**Confidence:** 5

**Review:**

A central component of multi-site studies is the correction for scanner/site biases. To correct for these biases, the article reviews the approach by Moyer et al. (2019), who proposes an unsupervised solution using invariant representations based on autoencoder. While the article is nicely written, the value add of this review over the original publications by Moyer et al.  is unclear.

---

### Official Review · AnonReviewer4 · 2020-03-13
**A clear review about an previously published original studies, although with limited originality**

**Rating:** 2
**Confidence:** 4

**Review:**

This paper reviewed a series of two studies by Moyer et al. 2018 and 2019, and provide an overview of the proposed method to create a latent representation of fMRI data and remove the effect of scanner variant.

Pros:
- Removing site effect from the data is an important topic. The original been reviewed here proposed an seemly effective way to achieve this task.
- In the discussion, the limitation of the methods is stated. Although it would be better to propose or discuss potential improvement to resolve the reduce the effect of "low information site".

Cons:
- I found the "overview" of the method to be too brief. For example, in figures:

    - In the main method Figure (1) describing the diagram of the network, details need to be added in the legend, e.g. the meaning of each term in the loss function (now only abbreviation is given, and the reader shouldn't need to refer to the original papers to find their meanings there)
    - In the main result paper, there is no description of what "Oracle" means. A brief description of the datasets used (Connectome 30/60, and Prisma 30/60) along with the meaning of the numbers comes along with It (30/60) should also be needed.

- The title is a bit misleading. The paper is mainly focusing on the overview of two previous papers, rather than an overview of the "scanner invariant representations" in more general sense, which is further limited by the lack of method comparison with other state-of-the-art methods used in papers mentioned in the introduction. Currently, only a comparison with a single baseline method which uses "template-based methods" which I found to be lacking.
- I find the originality and novelty is a bit missing in this study.

---

### Official Review · AnonReviewer3 · 2020-03-13
**A brief summary of the evaluation of a method for invariant representation of diffusion MRI**

**Rating:** 3
**Confidence:** 3

**Review:**

An evaluation of a method to eliminate the cross-site effect in diffusion MRI images is presented.
Pros:
- The work is framed within the state of the art and the current needs of the field,
- it seems promising
- The method proposed by the authors faces with relative success problems of high complexity

Cons:
- The description is so brief that it is complicated to evaluate how novel the work really is.
- The comparison with more methods is missing, especially because the work is presented as "Overview of Scanner Invariant Representations" and there is only one comparison.
- The authors should include some extra experiments that prove how invariant the representation achieved with the method really is. Personally, I would like to see two kind od comparisons:
    - Compare the usual diffusion MRI measures obtained after using the proposed method (or the methods shown if available) with the measures corrected using methods focused on the summary statistical level [1,2].
    - Use a trained classifier to differentiate between sites, after supposedly eliminating the effect with the method(s). Similar to how it is done in this paper [3]


[1] Jean-Philippe Fortin, Drew Parker, Birkan Tunc, Takanori Watanabe, Mark A Elliott, Kosha Ruparel, David R Roalf, Theodore D Satterthwaite, Ruben C Gur, Raquel E Gur, et al. Harmonization of multi-site diffusion tensor imaging data. Neuroimage, 161: 149–170, 2017.
[2] Artemis Zavaliangos-Petropulu, Talia M Nir, Sophia I Thomopoulos, et al. Diffusion MRI indices and their relation to cognitive impairment in brain aging: The updated multi- protocol approach in ADNI3. bioRxiv, page 476721, 2018.
[3] Glocker, B., Robinson, R., Castro, D. C., Dou, Q. & Konukoglu, E. Machine Learning with Multi-Site Imaging Data: An Empirical Study on the Impact of Scanner Effects. in Medical Imaging Meets NeurIPS 1–5 (2019).

---

### Meta-Review · Area_Chair1 · 2020-03-26
**MetaReview of Paper288 by AreaChair1**

**Rating:** 3

**Metareview:**

All reviewers agree on the relevance of the work. Most major concerns are related to the fact that the submission is a review of recently published work, but this is actually a format encouraged by the conference.

**Paper Type:**

methodological development

---

### Decision · Program_Chairs · 2020-04-11

Accept